# Commonly Assessed Markers in Childhood BCP-ALL Diagnostic Panels and Their Association with Genetic Aberrations and Outcome Prediction

**DOI:** 10.3390/genes13081374

**Published:** 2022-07-31

**Authors:** Jan Kulis, Łukasz Sędek, Łukasz Słota, Bartosz Perkowski, Tomasz Szczepański

**Affiliations:** 1Department of Pediatric Hematology and Oncology, Medical University of Silesia in Katowice, ul. 3 Maja 13-15, 41-800 Zabrze, Poland; slota.lukasz2392@gmail.com (Ł.S.); bartosz.perkowski@sum.edu.pl (B.P.); tszczepanski@sum.edu.pl (T.S.); 2Department of Microbiology and Immunology, Medical University of Silesia in Katowice, ul. Jordana 19, 41-808 Zabrze, Poland; lsedek@sum.edu.pl

**Keywords:** acute lymphoblastic leukemia, flow cytometry, *ETV6::RUNX1*, *KMT2A*, *TCF3::PBX1*, hyperdiploidy, hypodiploidy

## Abstract

Immunophenotypic characterization of leukemic cells with the use of flow cytometry (FC) is a fundamental tool in acute lymphoblastic leukemia (ALL) diagnostics. A variety of genetic aberrations underlie specific B-cell precursor ALL (BCP-ALL) subtypes and their identification is of great importance for risk group stratification. These aberrations include: *ETV6::RUNX1* fusion gene, Philadelphia chromosome (*BCR::ABL1* fusion gene), rearrangements of the *KMT2A*, *TCF3::PBX1* fusion gene and changes in chromosome number (hyperdiploidy and hypodiploidy). Diagnostic panels for BCP-ALL usually include B-cell lineage specific antigens: CD19, CD10, CD20, maturation stage markers: CD34, CD10, CD38, TdT, IgM and other markers useful for possible genetic subtype indication. Some genetic features of leukemic cells (blasts) are associated with expression of certain antigens. This review comprehensively summarizes all known research data on genotype-immunophenotype correlations in BCP-ALL. In some cases, single molecules are predictive of particular genetic subtypes, i.e., NG2 with *KMT2A* gene rearrangements or CD123 with hyperdiploidy. However, much more information on possible genotype or prognosis can be obtained with wider (≥8-color) panels. In several studies, a quantitative antigen expression scale and advanced statistical analyses were used to further increase the specificity and sensitivity of genotype/immunophenotype correlation detection. Fast detection of possible genotype/immunophenotype correlations makes multicolor flow cytometry an essential tool for initial leukemia diagnostics and stratification.

## 1. Introduction

Acute lymphoblastic leukemia (ALL) is the most frequent childhood neoplasm, accounting for more than 20% of all childhood cancers. It develops by malignant transformation of immature lymphocyte precursors in the bone marrow (BM) or thymus [1,2]. Of the two types of ALL, the one that originates from B-cell lineage—B-cell precursor ALL (BCP-ALL) is more common than the T-cell ALL (T-ALL) and accounts for > 80% of childhood ALL [3]. In recent years, the therapy effectiveness has been improved and the treatment protocols have been adjusted based on risk stratification resulting in raising the survival rates up to 80–90% [1,4]. However, relapses and therapy related complications still occur in 20% of cases [5].

Immunophenotype characterization with the use of flow cytometry (FC) is an essential tool for ALL diagnostics. It allows determination of the ALL subtype—one of the characteristics associated with disease outcome [1,2,6,7,8]. Diagnostic panels for BCP-ALL should include B-cell lineage specific antigens (CD19, CD10, CD20) and other markers useful for ALL subclassification and indicating clinically important abnormalities [9]. EuroFlow consortium have proposed an optimized diagnostic panel for BCP-ALL that consists of maturation stage markers: CD34, CD20, CD10, CD38, Terminal deoxynucleotidyl transferase (TdT) and IgM (heavy M/μ immunoglobulin chain); leukemia associated immunophenotype (LAIP) markers: CD33, CD13, CD9, CD81, and markers useful for possible genetic subtype indication: CD15, CD66c, CD123, NG2 [10,11]. Based on expression of CD10, IgM and cyIgK/cyIgL (cytoplasmic Kappa/Lambda light chain) the European Group for the Immunological Characterization of Leukemias (EGIL) identifies three BCP-ALL subtypes: BI - BIII (Table 1). The most mature B-IV subtype is now recognized as leukemic variant of Burkitt lymphoma—mature B-ALL [12,13].

In addition to immunophenotyping, a variety of genetic aberrations underlie the specific BCP-ALL subtypes and their identification is of great importance for risk group stratification. Quick detection of potential genetic aberrations is beneficial for the patients, as it enables appropriate therapeutic decisions to be taken [3,6,14]. While flow cytometry as a technique is historically older than modern cytogenetic and molecular genetic methods, it is still considered very useful and has many advantages, such as lower cost and faster results availability (within the same working day). Some genetic features of leukemic cells (blasts) are associated with expression of certain antigens. Therefore, immunophenotyping of blasts can provide information about genotype before the results of cytogenetic and/or molecular tests are obtained [6,10,15].

Commonly assessed genetic aberrations underlying BCP-ALL that have evidenced prognostic significance include:The *ETV6::RUNX1* gene fusion (previously known as *TEL-AML1*) is a result of chromosomal translocation t(12;21)(p13;q22) that is present in 22–25% of BCP-ALL, and correlates with good outcome in children [3,14,15,16,17,18,19,20].Philadelphia chromosome is the result of a reciprocal translocation t(9;22)(q34;q11) that results with a fusion gene named *BCR::ABL1* and is associated with poor prognosis for BCP-ALL patients [3,19,21,22].Philadelphia chromosome (Ph)-like ALL or *BCR::ABL1*-like ALL (Ph-positive ALL without *BCR::ABL1* fusion protein) is a subgroup of patients that do not have the *BCR::ABL1* fusion protein expressed from the Philadelphia chromosome but have a gene-expression profile similar to that of patients with *BCR::ABL1* ALL [19,23,24].Rearrangements of the *KMT2A* (previously named *MLL*) gene as a result of translocations within q23 region of chromosome 11 (t(11q23)) [6]. These aberrations are associated with an adverse clinical course of BCP-ALL, particularly in infant ALL [3,19,25,26].*TCF3::PBX1* fusion gene produced by t(1;19)(q23;p13.3) translocation [3,6,19]. Initially associated with poor outcome but recent therapies are improving the patient’s prognosis. However, the risk of CNS relapse appears to be higher in this genetic subgroup [19,20].Hyperdiploidy and hypodiploidy as a result of increase or decrease in chromosome number. Hyperdiploidy patients tend to have more favorable outcome [3] in contrary to patients with hypodiploidy [19,27].

Based on these features, BCP-ALL can be divided into corresponding genotypic subtypes. BCP-ALL that cannot be characterized by any of the above aberrations is often referred to as “B-other” subtype [3].

This review highlights the importance and meaning of specific markers commonly used for BCP-ALL diagnostics focusing on prognostic role. Although there are many studies on genotype- immunophenotype correlations, the majority of these studies focus on a single or few antigens. We wanted to sum up all known research data and present it in comprehensive form. We also wanted to highlight the utility of flow cytometry in BCP-ALL diagnostics.

## 2. Materials and Methods

For this review, the databases (including PubMed, Embase, Web of Science) were systematically searched for all relevant reports. The last search update was on 1 July 2022. Search terms included “acute lymphoblastic leukemia,” “BCP-ALL,” “immunophenotype,” “correlation” and all of the markers included in the study independently. The review also included the references from the identified papers. Because of the varying details of the methodology among cited studies, this review does not describe the methodological details of every study. In flow cytometric immunophenotyping, the most commonly used parameter describing the expression of a marker is the percentage of positive cells. The standard cutoff to consider the expression of an antigen as “positive” is 20%. Therefore, if the source literature did not state otherwise, >20% expression was noted as “+” in Table 2 in the Results section. In some cases, the expression was defined using median of fluorescence intensity (MFI). MFI allows for more precise interpretation and defines the expression gradually (from “low” to “bright”). The majority of studies transform the expression data into easy-to-understand terms, such as “positive”, “negative” (noted as “+” or “−“, respectively), “low” or “bright” (Table 2). If the source literature contained figures with dot-plots, we used our own cytometry experience to define the expression if it was not already stated by the authors.

## 3. Results


**CD19**


CD19 is a B lineage specific marker first identified as B4 antigen. It has a homogenous expression on the surface of hematogones, mature B lymphocytes and neoplastic B-cells. CD19 is present in almost every BCP-ALL case; it is a key component of both diagnostic and minimal residual disease monitoring panels in BCP-ALL [10,28]. Its expression is activated by the transcription factor Paired Box 5 (PAX5). The CD19 molecule is involved in B-cell receptor signaling (BCR). Its function is crucial for B-cell development and it may also have a critical role in the survival and maintenance of malignant human B lineage cells. Homogenous structure, high specificity to the B-cell lineage and the fact that it is a surface (easy to access) antigen make this molecule a good target for immunotherapy [28].


**CD10**


The CD10 antigen is expressed by normal B-cell precursor cells (also known as hematogones) found in bone marrow of healthy person. The highest levels of CD10 are visible on the earliest B-cell precursors. Following the maturation path, one can see that the CD10 expression slightly decreases on older hematogones and completely disappears on mature B-cells [9,29].

CD10 is also expressed on a subset of mature neutrophils but its expression is lower as compared to hematogones [9,29,30].

In BCP-ALL the CD10 is one of the most important markers that determine the BCP-ALL subtype. In the most frequent one—common-ALL—CD10 is strongly expressed on blast cells, exceeding the usual level of expression seen on normal hematogones. On the other hand, in pro-B subtype, blasts are CD10 negative [13,31].

CD10 negative subtype of BCP-ALL (pro-B) is often strongly correlated with rearrangements in the *KMT2A* gene—a feature considered as a poor risk factor [10,32,33].

In patients with the common BCP-ALL subtype with high expression of CD10 on blasts there is a higher chance for t(9;22)(chromosome Philadelphia) to occur [22,34]. In addition, these patients have a higher chance to manifest the t(12;21) *ETV6::RUNX1* fusion gene [33] but these relationships are considered as weak by most [32,35].


**CD66c**


CD66c is a heavily glycosylated carcinoembryonic antigen (CEA)-related glycoprotein that is involved in adhesion to another surface glycoprotein - E-selectin [36]. Its expression is usually limited to granulocytes and their precursors [37]; however, CD66c can also be expressed on blasts in BCP-ALL and is the most frequently expressed myeloid antigen in BCP-ALL. CD66c is associated with chromosome Philadelphia [21,32,37,38,39]. There are also some reports of CD66c positive cases in patients with hyperdiploidy [32,33]. Commonly assessed in BCP-ALL genetic aberrations: *ETV6::RUNX1* fusion gene, *PBX* and *KMT2A* rearrangement are more frequent in patients without CD66c expression. Therefore, with exclusion of t(9;22), high expression of CD66c could be considered as a good prognosis factor [32,38]. An example of overexpression of CD66c on blast cells is shown on Figure 1 as well as aberrant overexpression of NG2, TSLPR and CD123 molecules described below.


**NG2**


The neuron-glial antigen 2 (NG2) molecule was first discovered on oligodendrocyte progenitor cells in the mammalian central nervous system [25,26,40]. NG2 is often expressed on the surface of blast cells in pro-B (CD10 negative) BCP-ALL subtype. Studies show that NG2 expression is strongly associated with *KMT2A* rearrangement (Figure 1). Using the monoclonal antibody for cell surface NG2 recognition in a BCP-ALL immunophenotyping diagnostic panel can predict *KMT2A* before obtaining cytogenetic results [10,26,32,33]. Accuracy values for predicting *KMT2A* rearrangement reported by Emerenciano et al. vary from 0.67 to 0.93 for qualitative methods and 0.72–1.00 for quantitative methods, depending on time from sample collection to staining and testing [25]. Schwartz et al., calculated specificity and sensitivity equal as 0.89 (both) and positive prediction value as 0.93 for NG2 on a group of 120 adult patients [26].


**TSLPR/CRLF2**


Overexpression of thymic stromal lymphopoietin receptor (TSLPR) on BCP-ALL blasts is the result of the mutation in the gene encoding the cytokine receptor-like factor 2 (CRLF2) molecule—CRLF2 gene (Figure 1). This mutation leads to activation of STAT3 and STAT5 pathways mediated by JAK2 and has an impact on cell proliferation and development in the hematopoietic system [41,42]. This molecule is associated with Ph-like ALL genotypic subgroup [24]. TSLPR expression is correlated with IKZF1 deletion and is associated with poor prognosis for BCP-ALL patients. Patients with high expression of TSLPR are candidates for targeted therapy using inhibitors of molecules involved in JAK/STAT pathway [19,42,43].


**CD123**


CD123 molecule, also known as the interleukin-3 receptor alpha chain (IL-3Rα) is normally expressed by plasmacytoid dendritic cells. Various studies have shown that CD123 is often aberrantly overexpressed on BCP-ALL blasts in comparison to normal B cells (Figure 1) [44,45]. Therefore, CD123 is useful in monitoring the minimal residual disease (MRD). High expression of CD123 on blasts is known to be strongly correlated with hyperdiploid karyotypes [1,33,45,46]. The sensitivity and specificity of high CD123 expression for hyperdiploidy are 81.5% and 91.5%, respectively, with a positive predictive value of 78.6% [45]. Moreover, low CD123 is associated with *ETV6::RUNX1* genotype [45]. Additional studies have shown that CD123 is also associated with t(9;22)—*BCR::ABL1* genotype and that the CD123 molecule could also be a potential target for the immunotherapy in some cases [44].


**CD9**


CD9 belongs to the transmembrane tetraspanin 4 (TM4) protein family. Its biological functions include regulation of apoptosis, adhesion and motility. This molecule is expressed by B-cell precursors, mature lymphocytes, monocytes, megakaryocytes and other cells [14,47].

Some studies have shown that CD9 positive BCP-ALL cases have greater tumorigenic potential and drug resistance. It has been also observed that low or lack of CD9 expression is correlated with *ETV6::RUNX1* genotype—a molecular subtype with generally favorable prognosis [6,14,32,47]. Tsagarakis et al. have concluded that expression of CD9 shows direct correlation with *BCR::ABL1*, *KMT2A* (also confirmed by others [32]) and *TCF3::PBX1* genotypes [6].


**CD38**


Most characteristically, the highest expression of CD38 is observed on plasma cells [10,48]. CD38 is also strongly and consistently expressed by B-cell precursors. Other cell populations in bone marrow have a heterogenous and lower expression of this marker [10]. BCP-ALL blasts have often lower and heterogenous expression of CD38 which is useful in distinguishing these populations in MRD assessment [7,32,49].

CD38 has been found useful in outcome prediction. According to Chulian et. al., high CD38 expression shows significant direct correlation with the presence of *ETV6::RUNX1* gene fusion and significant inverse correlation with hyperdiploid karyotype. Lower levels of CD38 could be associated with a worse outcome in terms of survival [1]. However, other studies show that high CD38 expression among Ph negative patients is an indicator for early relapse risk [50].


**CD45**


CD45 antigen (the leukocyte common antigen—LCA) is a glycoprotein with a heavily glycosylated extracellular domain a transmembrane segment and a cytoplasmic fragment with tyrosine phosphatase activity. This molecule belongs to the protein tyrosine phosphatase (PTP) family. CD45 is a regulator of T-and B-cell-receptor signaling. CD45 is expressed on all cells in bone marrow except erythroblasts, erythrocytes and platelets, with the strongest expression on lymphocytes and moderate on neutrophils [4,36].

BCP-ALL blasts have often dimmer expression of CD45 than mature lymphocytes [51]. CD45 low and negative patients tend to have more favorable outcomes [51,52] and high expression is associated with worse event-free survival rate [4]. Studies show that low and negative CD45 can be associated with hyperdiploidy [11] and t(9;22)—*BCR::ABL1* genotype [32].


**CD20**


CD20 is a lineage specific antigen of B-cells with increasing expression level along their maturation from the lowest on B-cell precursors to the highest on mature B-lymphocytes [9,53].

CD20 is involved in B cell activation and regulation of their growth. This molecule also acts as a cell membrane calcium channel [36].

Strong expression of CD20 is more typical for B-cell lymphomas and Burkitt lymphomas than ALL [9,54]. In BCP-ALL blasts have often only partial or low expression of CD20 which, however, may increase during the treatment (particularly in the steroid therapy stage) [36].

In adult BCP-ALL cases, the expression of CD20 is associated with poor prognosis and higher risk of relapse. However, in pediatric BCP-ALL there is no clear evidence of CD20 impacting the disease outcome [54]. Still, some researchers report that (over)expression of CD20 could be associated with t(9;22)—*BCR::ABL1* genotype and can be useful to exclude *KMT2A* gene rearrangement even when expressed at low levels [32].

A chimeric anti-CD20 monoclonal antibody—Rituximab targets the CD20 molecule and is useful for treatment CD20 positive BCP-ALL patients [36].


**CD24**


CD24 is a glycosylphosphatidyl inositol (GPI)-linked sialoglycoprotein and is encoded by a gene that is mapped to the long arm of chromosome 6. CD24 appears during B-cell maturation in BM at the stage of pre-B-I cells and is upregulated until mature B-cell stage [55]. The expression of CD24 disappears when B-cells transform to plasma cells. Apart from B-cells, the expression of CD24 is observed on neutrophils. CD24 is also a sensitive marker of most mature B-cell malignancies; however, its expression may be variable [56,57]. Studies show that CD24 is usually expressed on blast cells despite the patient’s genotype [32] but lack of CD24 can be associated with *KMT2A* gene rearrangement [26,32].


**CD22**


CD22 is another B-cell-associated marker [58], expressed from the early stages of B-cell (pro-B-cell) maturation in BM; it precedes CD19 which appears at the stage of pre-B-I cells. Its expression is usually dim/moderate on all stages of hematogones and is upregulated on peripheral mature B-cells [53]. Leukemic blasts in BCP-ALL usually exhibit low expression levels, similar to those of normal hematogones [9,13]. This antigen became already a widely used target in patients undergoing immunotherapy. Moxetumomab (anti-CD22 immunotoxin) is an example of a promising monoclonal antibody-based drug for patients with a poor response to standard chemotherapy-based treatment [58]. Other anti-CD22 antibodies that can be useful in immunotherapy are Epratuzumab and Inotuzumab Ozogamycin [36]. Studies show that low positive expression of CD22 can be associated with *BCR::ABL1*, *TCF3::PBX1* and *ETV6::RUNX1* genotypes, bright with hyperdiploidy [32]. CD22 is usually absent in cases with *KMT2A* gene rearrangement [32,59].


**CD13**


A marker normally associated with myeloid cell lineages, involved in the alteration of target cell specificity and acting to activate or inhibit bioactive proteins. CD13 also controls the chemotactic response of neutrophils to IL-8 and is involved in tumor migration as it supports cell adhesion [30].

Blast cells in BCP-ALL can have aberrant expression of molecules specific to other than B-cells lineage such as CD13, which is normally associated with myeloid lineage or early hematopoietic stem cells [60,61]. CD13 positive BCP-ALL cases are more likely to have *ETV6::RUNX1* or *BCR::ABL1* genotype [16,22,34,39,62]. Hrusak et al. suggest that CD13 is usually absent in *TCF3::PBX1*, *KMT2A* and hyperdiploidy genotypes [32].


**Cytoplasmic IgM (cIgM)**


As a component of normal B-cell receptors, cytoplasmic expression of this heavy immunoglobulin chain receptor starts at the stage of late pre-B-I cell and is upregulated along further maturation of B-cells in BM. Immature B-cells exhibit at least partial surface expression of IgM while in mature B-cell IgM is only present on the cell surface membrane. Expression of cytoplasmic IgM in BCP-ALL determines the pre-B subtype [13,31,63,64]. Studies show that cIgM can be associated with *TCF3::PBX1* gene rearrangement and is usually absent in patients with other common genetic aberrations [6,10,15,32].


**CD15**


CD15 is expressed by monocytes and granulocytes and is involved in carbohydrate interactions and phagocytosis. CD15 is expressed in high levels on mature granulocytes [30,65,66].

CD15 can also be detected on the surface of BCP-ALL blasts in pro-B (CD10 negative) subtype. Therefore, like NG2, this molecule is associated with *KMT2A* gene rearrangement and is often absent in other genetic subtypes [6,10,32].


**CD33**


CD33 is a transmembrane sialic acid binding immunoglobulin-type lectin (SIGLEC) receptor which is involved in negative selection of human self-regenerating stem cells [30,67]. Other known functions of CD33 include: apoptosis induction and cytokine secretion modulation [36]. CD33 is expressed on mature myeloid cells and hematopoietic progenitor cells (but with low or negative expression on early normal hematopoietic stem cells), with the highest expression on mature monocytes. Its expression is commonly found on blasts in AML, however it may also be aberrantly expressed by blasts in BCP-ALL [10,32,61,67,68].


**TdT**


Terminal deoxynucleotidyl tranferase (TdT) is an intracellular and intranuclear DNA polymerase that catalyzes the template-independent addition of deoxynucleotides to the 3’-hydroxyl terminus of oligonucleotide primers. TdT is strongly expressed in lymphoid precursor cells (both T and B-cells) and usually is present on blast cells in ALL, with the exception of very immature subtypes of T- and B-lineage ALL (ETP-ALL and pro-B-ALL, respectively) [13]. Low expression of TdT can be associated with *KMT2A* gene rearrangement [33]. This marker is also useful in distinguishing ALL from mature B-cell malignancies (including Burkitt lymphoma, BL) which typically are TdT-negative [9,69,70].


**CD81**


CD81 is an integral surface membrane protein with 4 transmembrane domains, is involved in carrying out signal transduction and is crucial for B-cell development and the humoral response. CD81 has the highest expression on normal lymphocytes. On the other hand, CD81 is expressed with lower intensity and greater variability on granulocytes, monocytes, basophils, and plasmacytoid dendritic cells (PDCs). CD81 is not expressed on erythrocytes and platelets. CD81 is aberrantly decreased in B lymphoblastic leukemia blasts, particularly in CD34-positive cases [10,49,71]. High expression of CD81 lowers the chance for *ETV6::RUNX1*, *KMT2A* and hyperdiploidy to occur [33].


**CD34**


In the bone marrow, CD34 is primarily expressed in stem cells and their malignant counterparts. This marker is a cell surface glycoprotein that functions as a cell–cell adhesion factor [69]. B-cell hematogones have CD34 on their surface at the earliest stage of development. It is also commonly present on blasts in BCP-ALL but the expression is often heterogenous [10,29,62,72]. Lack of CD34 can be associated with the rearrangements of the *KMT2A* gene and *TCF3::PBX1* fusion gene [6,32,33].

The table does not include CD19 antigen because it is expressed in virtually every case of BCP-ALL. For some markers the prognostic significance cannot be determined. There are no reports about TSLPR molecule being associated with genetic aberrations included in the table. However, it is associated with Ph-like ALL and unfavorable outcomes.

**Table 2 genes-13-01374-t002:** Immunophenotypic features of studied genetic subtypes.

Antigen	*ETV6::RUNX1*	*BCR::ABL1*	*KMT2A*	*TCF3::PBX1*	Hyper-Diploidy	Prognosis if +	EGIL Subtype if +
CD10	+ [32,33,35,47,62]	+ [6,22,32,34]	− [10,13,32,33,59]	+/− [13,32]	+ [11,32]	Good [26]	B-II [13,31]
CD66c	− [13,32,38,73]	+ [13,21,32,37,38,39]	− [6,32,38]	− [32,38]	+ [11,32,33]	Inconclusive	Irrelevant
NG2	− [32]	− [32]	+ [10,25,26,32,33]	− [32]	[32]	Poor [3,25,26]	B-I [10,26,32]
TSLPR	ND	ND	ND	ND	ND	Poor [24,42,43]	Irrelevant
CD123	Low + [45]	+ [44]	+ [6]	− [6]	+ [1,33,44,45,46,74]	Good [3,45]	B-II [45]
CD9	− [6,13,14,32,47]	+ [6]	+ [6,32]	+ [6,32]	ND	Poor [14,47]	Irrelevant
CD38	+ [1,33]	− [6,34]	+ [6]	+ [6]	− [1]	Inconclusive [1,50]	Irrelevant
CD45	Low+ [16,32,33,62]	−/Low+ [32]	Low+ [32]	Low + [32]	− [11,32,53]	Poor [4,51,52]	Irrelevant
CD20	− [13,16,32,62]	+ [32]	− [6,13,32,59]	Low + [13,32]	Low+ [32]	Inconclusive [39,54]	Irrelevant
CD24	+ [13,32]	+ [13,32]	− [13,26,32,33]	+ [32]	+ [32]	Inconclusive	B-IV [56,57]
CD22	Low+ [32]	Low+ [32]	− [32,59]	+ [32]	+ [32]	Inconclusive	All [13,31]
CD13	+ [13,22,32]	+ [16,32,34,39,62]	− [32]	− [32]	− [32]	Inconclusive	Irrelevant
cIgM	− [32]	− [32]	− [6,32]	+ [6,10,15,32]	− [32]	Poor	B-III [13,31,63,64]
CD15	− [32,62]	− [6,32,39]	+ [6,10,13,26,32,59]	− [32]	− [32]	Poor	B-I [13]
CD33	+ [13,32]	+ [13,22,39]	− [32]	− [32]	− [32]	Poor [1]	Irrelevant
TdT	+ [13,32]	+ [13,32]	Low+ [33]/ + [13,32]	+[32]	+[32]	Inconclusive	B-I, II, III [13,31]
CD81	Low+/−[33]	ND	− [33]	ND	− [33]	Good	Irrelevant
CD34	Low+ [33]/ + [6,13,32,62]	+ [6,13,34]	− [6,32,33]	− [6,13,32]	+ [32,33]	Inconclusive	Irrelevant

## 4. Discussion

Single antigen expression can rarely predict a molecular genetic subtype. Only some exceptional molecules appear to have strong prediction value. For example, NG2 antigen was associated with *KMT2A* gene rearrangement in one of the first reports about immunophenotype-genotype correlations. It was confirmed by other researchers in later studies and is still considered a useful feature for *KMT2A* rearrangement prediction [10,25,26,32]. Overexpression of CD123 is another example of single molecule that can be strongly associated with genotype. Many studies have confirmed a decent sensitivity and specificity for this marker to predict the hyperdiploidy [1,44,45,74].

Some markers have been studied more frequently than others and are well known to be associated with specific genotypes. CD66c [1,21,32,37,38,39] and CD13 [16,32,34,39,62], for example, are associated by many with *BCR::ABL1*. Lack of CD9 is often correlated with *ETV6::RUNX1* genotype [6,14,32,47]. Expression of a well described marker such as CD10 can be instantly associated with good prognosis. Similarly, NG2 can be instantly identified as a poor prognosis feature. On the other hand, markers like CD66c can be associated with few genotypes that are either favorable or not. Therefore, finding CD66c expression remains inconclusive in terms of prognosis (see Table 2 for details). Unfortunately, some markers, such as CD24 or CD33, have been studied only rarely or were only mentioned in a review paper [32].

Instead of a single marker analysis, a whole immunophenotype of blast cells should be assessed to obtain optimal genotype prediction values. Proposed immunophenotypes for corresponding genotypes can be found in Table 2. In majority of cases, the obtained results are consistent.

Different study groups have been using different methods for reporting a “+” expression. The usual cutoff for considering a population to have expression of given marker is set to 20%. However, FC immunophenotyping is difficult to standardize. Using different reagents on different machines can produce inconsistent results. Considering how old are some studies reported here and how the technology has developed over time, the 20% cutoff appears to be prone for misinterpretation and neglects the expression intensity. For antigens that have a heterogenous expression, using the percentage may be the correct way but overall, using the median of fluorescence intensity (MFI) better represents the actual expression level. Furthermore, studies presented in this paper have been conducted on various groups of patients in terms of age.

A majority of papers describe the expression of antigens measured qualitatively (as “+”, low or negative) and the prediction is based on one-dimensional statistical analysis. In a few instances, specificity and sensitivity of the molecular aberration prediction is provided. Given the mutually exclusive nature of the four major cytogenetic translocations, a straightforward correlation study might be sufficient in some cases. There are a few review-type studies collecting the genotype/immunophenotype correlation data and clinical outcome results, either focusing on one marker or a whole immunophenotype. It is worthwhile mentioning that a study by Hrusak et al., from 2002, provides highly detailed information that was not provided by others.

There are only few studies with more advanced statistical analysis [25,26,33,45] and with expression level measured quantitatively (with gradual expression assessment) [6,33,34,62].

Taberneo et al. used MFI and coefficient of variation (CV) to assess the intensity and homogeneity of expression of antigens. They presented the scoring system and concluded that a specific pattern of expression (homogenous + CD34, homogenous + CD10, intermediate or low CD38 and intermediate + CD13) is useful for predicting the *BCR::ABL1* in adult BCP-ALL patients [34].

De Zen et al. proposed an original approach where the absolute levels of antigen expression were transformed from fluorescence channels into “molecules equivalent to soluble fluorochrome” (MESF). This method permitted accurate results with coefficient of variation (CV) and intensity of expression taken to account [62].

Tsagarakis et al. proposed the scoring system (SS) for predicting four genetic aberrations in patients with BCP-ALL. The system is based on the percentage of expression of particular antigens. The points are added or subtracted for expression of given marker and the calculated score is used to accurately predict a respective genetic rearrangement. The described method appears to be highly specific and significantly sensitive [6].

Kulis et al. developed a system based on the median of fluorescence intensity with a gradual antigen expression scale. Machine learning methods were applied for the genotype/immunophenotype correlation analysis. This approach enabled the researcher to expand and specify the meaning of particular expression levels of markers. The study also made an application tool for prediction studied aberrations, open to use by other researchers. Specific antigenic patterns were proposed that were strongly associated with respective aberrations. Due to the high complexity of immunophenotype description, most features cited in the results section (Table 2) were simplified for the sake of readability [33].

## 5. Conclusions

Since the development of multidimensional flow cytometry, much work has been undertaken to characterize the immunophenotypic features of BCP-ALL genetic categories. A wide panel of markers routinely applied for BCP-ALL diagnosis can provide substantial information about patient’s prognosis and the likely course of a disease. Overall, well established knowledge about genotype/immunophenotype correlations makes the multicolor flow cytometry an essential tool for initial leukemia diagnostics.

## Figures and Tables

**Figure 1 genes-13-01374-f001:**
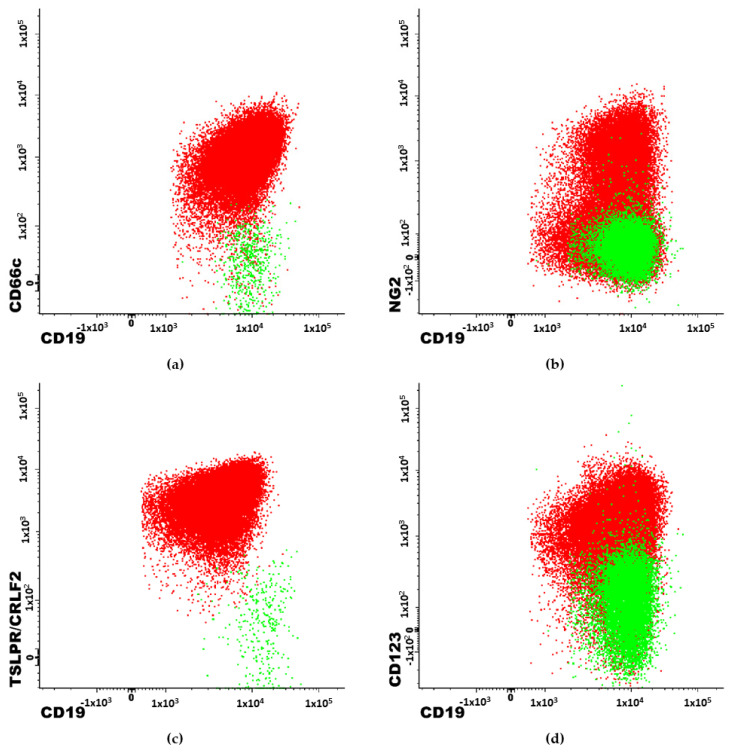
2-dimendional dot-plots obtained with Flow Cytometry. Examples of immunophenotypic features of patients with BCP-ALL. Red population represents blast cells. Highlighted in green are the normal B-cells. (**a**) patient with Philadelphia chromosome. (**b**) patient with *KMT2A* gene rearrengement. (**c**) patient with TSLPR/CRLF2 antigen overexpression (Ph-like associated feature). (**d**) patient with hyperdiploidy.

**Table 1 genes-13-01374-t001:** Immunophenotypic subtypes of BCP-ALL according to European Group for the Immunological Characterization of Leukemias (EGIL) [12,13].

BCP-ALL Subtype	CD10	cyIgM	sIgM and/or cyIgK or cyIgL
B-I (pro-B) ALL	−	−	−
B-II (common) ALL	+	−	−
B-III (pre-B)	+	+	−
B-IV (mature)	−/+	+	+

cyIgM—cytoplasmic IgM; sIgM—surface IgM; cyIgK—cytoplasmic Kappa light chain; cyIgL—cytoplasmic Lambda light chain. Additional subtype of BCP-ALL—Transitional pre-B-ALL subtype is characterized by presence of surface and cytoplasmic IgM but lack of cyIgK or cyIgL light chains [13].

## Data Availability

Not applicable.

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
