# Peer review of "Commonly Assessed Markers in Childhood BCP-ALL Diagnostic Panels and Their Association with Genetic Aberrations and Outcome Prediction"

_genes, 2022, doi:10.3390/genes13081374_

Round 1

Reviewer 1 Report

In this review, the authors describe each of immunophenotypic markers used for of BCP-ALL diagnosis and comprehensively summarize all known research data on genotype-immunophenotype correlation in BCP-ALL.  The information is very useful for clinicians.  I have several minor comments.

1.       Table 1 and Figure 1 are not explained in the text. 

2.       On page 3, line 92: “can be” is duplicated.

3.       On page 3, line 102: “The” before “For this review” is not necessary.

4.       On page 3, line 105: “alle” should be “all”.

5.       On page 3, line 10: “de-scribe” should be “describe”.

6.       On page 10, in legend of Figure 1(d): “hiperdiploidy” should be “hyperdiploidy”.

7.       On page 11, line 379: What is “CV”?

Author Response

In this review, the authors describe each of immunophenotypic markers used for of BCP-ALL diagnosis and comprehensively summarize all known research data on genotype-immunophenotype correlation in BCP-ALL.  The information is very useful for clinicians.  I have several minor comments.

The authors thank the reviewer for recognizing the usefulness of this research.

  1. Table 1 and Figure 1 are not explained in the text.

The authors thank for this remark. New information on Table 1 was included in the Introduction section. The explanation of Figure 1 was added in the Results section of the revised manuscript.

  1. On page 3, line 92: “can be” is duplicated.
  2. On page 3, line 102: “The” before “For this review” is not necessary.
  3. On page 3, line 105: “alle” should be “all”.
  4. On page 3, line 10: “de-scribe” should be “describe”.
  5. On page 10, in legend of Figure 1(d): “hiperdiploidy” should be “hyperdiploidy”.

The authors thank for these remarks. The typing errors were corrected.

  1. On page 11, line 379: What is “CV”?

The authors thank for this remark. The explanation of the abbreviation was included in the revised manuscript.

Reviewer 2 Report

Congratulations to the authors for the chosen topic. It is a challenge to correlate multiple immunophenotypic and genotypic markers in pediatric  B-cell precursor ALL and to obtain objective results regarding their prognostic role.

Line 110 Although the methodology is briefly detailed and I understood the authors' explanations and the limitations they encountered at this point, a more detailed description of the methods is recommended.

Line 115 Considering the current importance of CD19, I recommend a more detailed description of this marker.

Author Response

Congratulations to the authors for the chosen topic. It is a challenge to correlate multiple immunophenotypic and genotypic markers in pediatric B-cell precursor ALL and to obtain objective results regarding their prognostic role.

The authors thank the reviewer for recognizing the challenge of this research.

Line 110 Although the methodology is briefly detailed and I understood the authors' explanations and the limitations they encountered at this point, a more detailed description of the methods is recommended.

The authors thank for the remark. The methodology section was expanded in the revised manuscript.

Line 115 Considering the current importance of CD19, I recommend a more detailed description of this marker.

The authors thank for the remark. The extended description of CD19 marker was included in the revised manuscript.